# Development and Psychometric Properties of the Spanish Social Stigma Scale (S3)

**DOI:** 10.3390/healthcare12131242

**Published:** 2024-06-21

**Authors:** José Germán Arranz-López, Jorge Pérez-Corrales, Francisco H. Machancoses

**Affiliations:** 1Child Psychiatric Referral Unit Institut Pere Mata, Carretera Institut Pere Mata, 6, 43206 Reus, Spain; arranzj@peremata.com; 2Research Group of Humanities and Qualitative Research in Health Science (Hum&QRinHS), Department of Physical Therapy, Occupational Therapy, Physical Medicine and Rehabilitation, Universidad Rey Juan Carlos, Avenida Atenas, s/n, 28922 Alcorcón, Spain; 3Predepartamental Unit of Medicine, Facultat de Ciencies de la Salut, Universitat Jaume I de Castellón, Avda. Sos Bainat, s/n, 12071 Castelló de la Plana, Spain; herrerof@uji.es

**Keywords:** social stigma, mental health, mental disorder, scales

## Abstract

(1) Background: Mental health problems are associated with negative connotations that may lead to discrimination and rejection of people diagnosed with mental disorders. The present study aimed to develop and validate a new scale (the Spanish Social Stigma Scale—S3) to assess the current level of social stigma in the general Spanish population. (2) Methods: The assessment tool was developed after reviewing the items of existing tools that represent the most appropriate indicators for the assessment of social stigma. A review was performed by volunteer subjects and by a group of experts in the field, based on the participation of 563 respondents to a survey. (3) Results: The confirmatory factor analysis revealed that the developed tool fits with the factors that determine the level of social stigma and shows good internal consistency (χ^2^_SB_ = 412.0321, gl = 293, *p* < 0.01; BBNNFI = 0.922; CFI = 0.930; IFI = 0.931; RMSEA = 0.028 [0.022, 0.035]). (4) Conclusions: The S3 is useful for assessing knowledge, attitudes, and behavior towards people diagnosed with a mental disorder. This tool may be used for the identification and development of mechanisms necessary for the reduction of social stigma in the general population.

## 1. Introduction

Stigma towards mental health issues is an ongoing issue in current society [1,2]. Indeed, mental health constitutes a major concern for public health, not only due to its prevalence in the population, the burden of disease, the situation of disability that it generally causes [3,4] and high suicide rates [5], but also because of the rejection it causes in the general population [6,7,8].

This leads to the discrimination and rejection of people diagnosed with mental disorders, their family members, people close to them, and the mental health resources they use, among others [7,9].

The detrimental image of people with mental health problems generates unfavorable situations regarding their quality of life that affect other factors such as income, interpersonal relationships, community participation, access to training, and employment [10]. These aspects, in turn, reinforce the negative image of people diagnosed with mental disorders [6,7,8]. Moreover, the social stigma towards mental illness interferes with the ability to access the specific health and/or social resources available to those who begin to suffer from mental health problems and their families [7,11]. Concretely, people may fail to identify with the pejorative image of the disorder or express fear of being labeled as “mentally ill” [7,11]. This refusal to be stigmatized can lead individuals with mental health problems to avoid or ultimately refuse seeking professional help [12].

It is important to note that social stigma, like mental health, is influenced by culture and the particular historical period [2,7]. It is defined as “an attribute or label that deeply discredits the person or group of people, undervaluing the stigmatized person and reducing their social value” [13]. In addition, it is based on three distinct dimensions: knowledge, attitudes, and behavior, directly related to the concepts of stereotype, prejudice, and discrimination [7,8,14,15,16]. These dimensions offer an approach to the social stigma phenomenon based on the Knowledge-Attitude-Behavior (KAB) model, a cognitive sphere called stereotype, which is the set of beliefs about the generally negative attributes assigned to a group; an emotional sphere or prejudice, understood as the negative affection or evaluation towards the group; and finally, a behavioral sphere or social discrimination, based on actions aimed at the lack of equality in the treatment of the stigmatized person or group of people [11,12,13,14,15,16,17].

In turn, the stigmatization process unfolds over four distinct stages [8]. The first is in which people distinguish and label a difference or characteristic as being socially salient, which, in turn, initiates a process of social differentiation into groups. The second is that these differences or labels are associated with detrimental characteristics, thus creating a negative stereotype that is applied to each component of the group. In the third phase, the labeled and stereotyped individuals are identified as distinct within the social group, creating a separation. Finally, the differentiated group is devalued, deprived of an identity different from the stereotyped one, and disadvantaged in terms of social factors. The latter is related to the lack of social participation in family, friendships, romantic relationships, or community contexts. Moreover, economic factors are notable, related to the absence of training and employment opportunities, financial income, and housing, among others, which lead to a worsening of mental health problems and overall health [2,17,18].

Different scales exist for the assessment of the different dimensions of social stigma in the general population in relation to people with mental health problems [11]. Factors frequently measured to assess social stigma include mental health literacy in different areas such as help-seeking, recognition, support, employment, treatment and recovery, and knowledge about illness, among others [19,20,21]. Also, community attitudes towards people with mental health diagnoses in relation to items that identify factors such as authoritarianism, benevolence, social restrictions, and ideologies regarding mental health are considered [22,23,24,25,26]. Finally, behaviors towards people with mental health diagnoses in areas of housing, work, social relationships, or factors such as level of danger, safety, and trust are also major influences on the behavior of people with mental health diagnoses [19,20].

The main objective of the present work was to develop and validate an updated tool for assessing the level of social stigma towards people with a mental health diagnosis in the Spanish population. The purpose of the development of this new scale, in addition to increasing the knowledge and study of stereotypes, social attitudes, and behaviors of social discrimination in the current population, is to provide relevant knowledge regarding the identification of the mechanisms necessary for the reduction of stigma from a structural and individual perspective of the problem, incorporating the three dimensions described above in a single measurement scale.

## 2. Materials and Methods

### 2.1. Study Design

The first step for the creation of the S3 scale, was to conduct a review of several existing scales (Table 1), to identify the different dimensions related to the stigmatization process, according to various authors, with the aim of designing a tool that would include this process in its entirety. Subsequently, after reviewing the existing assessment tools, a scale for the assessment of social stigma was developed following the three-dimensional model: knowledge or stereotype, cognition or attitude, and behavior or discrimination [1,14,15,27,28].

In the initial study, the scale was examined by three experts in the field (two mental health professionals and one researcher who was an expert in psychology research methodology) for the inter-rater assessment of the items, with the aim of evaluating the structure, content, and comprehension of the language used. To this end, their comments and evaluations of the different items were collected. Subsequently, a group of volunteers was asked to rate the different items in terms of their ability to evaluate the social stigma.

In this study, the three factors or dimensions that conform to the scale were used as primary variables: indicators of stereotypes, indicators of stigmatizing attitudes, and indicators of discrimination against people diagnosed with mental disorders. The secondary variables recorded were age, gender, marital status, autonomous community of residence, political ideology, level of education, existence of training related to mental health, economic situation, and proximity to a person or persons with a diagnosis of mental illness.

### 2.2. Spanish Social Stigma Scale (S3)

The assessment instrument initially consisted of a total of 30 items assessed by means of a Likert-type polytomous scale, with scores between 1 and 5, based on the level of agreement of the respondent and considering that the higher the score, the higher the level of stigma. For its interpretation, a total score is calculated as well as a score for each dimension, i.e., stereotype, attitude, and discrimination (Table 2), which is subsequently transformed into centile scores for standardization and comparison with the remaining scales.

Thus, the stereotype dimension provides information on the degree of stereotypical thoughts or erroneous knowledge presented by the person. Stigmatizing attitudes are related to the affective dimension of social stigma. Finally, behavior towards others, or discrimination, identifies different actions based on rejection towards people with mental disorders.

### 2.3. Sample

The study population mainly consisted of 712 social network users who voluntarily agreed to participate in the study and who were residents of Spain, representing a broad group of the Spanish population in general. For this purpose, prior contact was established with different users with a large number of followers, with non-governmental organizations, and with different mental health care resources so that they could disseminate the survey link. Two weeks after beginning the dissemination of the survey and after eliminating incomplete surveys, a total of 563 respondents were obtained, and the responses were analyzed.

### 2.4. Data Analysis

First, a univariate descriptive analysis was carried out of both sociodemographic variables and of the main variables, to describe these variables in the sample using frequencies and percentages.

For the validation of the instrument, an internal consistency analysis was carried out using Cronbach’s Alpha, McDonald’s omega, and a confirmatory factor analysis (CFA) to determine whether the data fit the theoretical model used. For this purpose, the EQS 6.2 program was used [30]. The advantages of the EQS 6.2 software over others are that it introduces the robust Satorra–Bentler statistic [31], which enables the testing of hypotheses on relationships between latent variables and indicators, including the different interrelationships between them, when the assumptions of normality and heteroscedasticity are not met. The EQS also offers the Lagrange Multiplier Test, a procedure designed to study the need for restrictions in the model, both equality restrictions that may have been included and covariance not initially included and that should be counted as free parameters; and the Wald test for dropping parameters, and the Wald test to drop parameters, which makes it easier for us to choose together with “Cronbach’s Alpha if the element is eliminated” to reject the items that do not work correctly [32].

The adjusted model was determined by the following indexes: Robust Independent Model χ^2^ and Satorra–Bentler Scaled χ^2^; Non-Normed Fit Index, Normed Fit Index, and Comparative Fit Index, which should take values greater than 0.9; and the Root Mean Square Error of Approximation, which should be less than 0.08. According to MacCallum et al. [33], RMSEA values between 0.06 and 0.08, together with other coefficients greater than or equal to 0.9, indicate an appropriate fit.

## 3. Results

### 3.1. Participant Characteristics

The sample consisted of 563 participants, with an average age of 35.81 years, mostly women (n = 381, 65.20%). The predominant political ideology in the work sample was progressive (n = 309, 65.20%). A total of 75.90% (n = 417) had university or postgraduate studies (university degree, master’s degree, or doctorate). It should be noted that 54.60% (n = 302) had specific training in mental health. In addition, 78.50% (n = 422) of the sample had had contact with at least one person with a mental health diagnosis during their lifetime (Table 3).

### 3.2. Validation of the Scale

The internal consistency of the total scale was α = 0.839 and Ω = 0.841, which is considered a very good internal consistency [34]. Regarding the dimension ‘Erroneous or Stereotyped Knowledge’, α = 0.659 and Ω = 0.655 are observed, observing that by eliminating item 1 this would increase to 0.681 (Table 3). As for the attitude dimension, an α = 0.660 is observed, and by eliminating item 28 and item 14, Cronbach’s Alpha increases to 0.682 and 0.707, respectively. Finally, in the discrimination dimension, α = 0.634 is observed, and by eliminating item 12, Cronbach’s Alpha increases to 0.733.

### 3.3. Construct Validity (Confirmatory Factorial Analysis)

To determine whether the proposed three-dimensional model fits the data, a confirmatory factor analysis was carried out. Considering the scale as unidimensional and bearing in mind the existence of a second order factor (stigma), and after the elimination of items 1, 4, 12, 20, and 23, these being some of those already shown to be deficient in the analysis of the consistency of the dimensions (Table 4), it was observed that the proposed model fits correctly, given the fit indices: χ^2^_SB_ = 412.0321, gl = 293 (*p* < 0.01), BBNNFI = 0.922, CFI = 0.930; IFI = 0.931; RMSEA = 0.028 [0.022, 0.035].

In addition to the fit indices and the saturations of the latent and observable variables, the EQS provides us with the Cronbach’s Alpha of the model, equaling 0.843 (Ω = 0.843), somewhat higher than the Alpha of the complete scale before the elimination of five items. When removing the items, we considered both the corrected item-test correlation tests carried out in the item analysis of the subscales and the indicators in the confirmatory factor analysis (Figure 1).

After the analysis and validation of the scale administered for the assessment of knowledge, attitudes, and stigmatizing behaviors in the Spanish population, the Spanish Social Stigma Scale (S3), and after the elimination of the five items that reduce consistency, the findings revealed that the instrument adjusts correctly to the proposed model, maintaining good overall internal consistency, and equally for each of the dimensions, making it valid for administration in future studies as a measure of the level of social stigma (Table 5).

### 3.4. Scale Explanation

This is a 25-item scale that includes three dimensions: Factor 1: knowledge (9 items), Factor 2: attitude (9 items) and Factor 3: discrimination (7 items). Positive and negative expressions are used (items 2, 4, 13, 16, and 24 are reverse scored). Each item is rated on a five-point Likert scale (5 = completely agree; 4 = mildly agree; 3 = neither agree nor disagree; 2 = mildly disagree; 1 = completely agree). The total and dimension scores are the sum of the scores for each item. A higher score infers a higher degree of stigma.

## 4. Discussion

The initial assumptions motivating this study on social stigma and the variables that may influence stereotypes, stigmatizing attitudes, and discriminatory behaviors that directly affect people with a mental health diagnosis largely coincide with the results obtained after the creation and validation of the Spanish S3.

Our data confirm the three-factor model of social stigma (knowledge, attitudes, and behavior), as stated by Michaels et al. [1], Reneses et al. [15], and Tekin and Outram [7], among others. These three factors are directly related to each other through a second-order factor (social stigma).

In addition, the current scale offers a global measure and a measure of the different spheres of social stigma, offering information on stereotypes, stigmatizing attitudes, and discriminatory behaviors and how these three dimensions impact a greater or lesser degree of social stigma.

There is currently no validated scale in Spanish that evaluates social stigma based on the KAB model or three-dimensional model. The available scales are based on other factors such as help-seeking, recognition, support, employment, treatment and recovery, and knowledge about illness, among others [19,20,21]. Other dimensions investigated are authoritarianism, benevolence, social restrictions, and ideologies regarding mental health [22,23,24,25,26].

In this study, it is important to consider the limitations associated with the sample characteristics. Firstly, there is a gender bias, as 68.90% of the participants were women. Another relevant bias is the political bias, given that the predominant ideology in the sample was progressive (65.20%). Likewise, the educational level of the sample (75.90% had university or postgraduate education) could influence attitudes toward social stigma, excluding individuals with lower educational levels. Furthermore, the fact that 54.60% of participants had specific mental health training could bias responses and attitudes. Lastly, prior contact with individuals diagnosed with mental health conditions (78.50% of the sample) could influence perceptions and attitudes toward stigma. In summary, while these findings are valuable, it is essential to acknowledge these limitations and consider how they may affect the generalizability of the results to the overall population.

## 5. Conclusions

Our tool offers a measure of three dimensions on a single scale, providing information on the degree of stereotyped thoughts or erroneous knowledge presented by the person, the affective dimension of HS, and behaviors of rejection towards people with a mental health diagnosis in both a unidimensional and global manner.

In addition, this new social stigma assessment measure provides a useful and rapid measure for its administration to the Spanish population. This is appropriate for the measurement of the three specific dimensions that explain social stigma, is based on the KAB model, and contains an adjusted number of items, unlike other previous assessment tools that are more extensive yet less comprehensive.

Improving our knowledge and understanding of the influence of social stigma towards people with a mental health diagnosis among the Spanish population should serve to detect and respond to current needs and break down the barriers that interfere with access and outreach to mental health associations and resources. Also, to reinforce the promotion of a holistic and adequate recovery of a person’s objectives in life, promote a full, meaningful, and satisfactory psychosocial and community rehabilitation for the person with a mental health diagnosis.

In addition, new studies are needed to expand the quality and number of studies on the importance of social stigma in the general population, and particularly among people diagnosed with mental disorders, or “self-stigma”.

## Figures and Tables

**Figure 1 healthcare-12-01242-f001:**
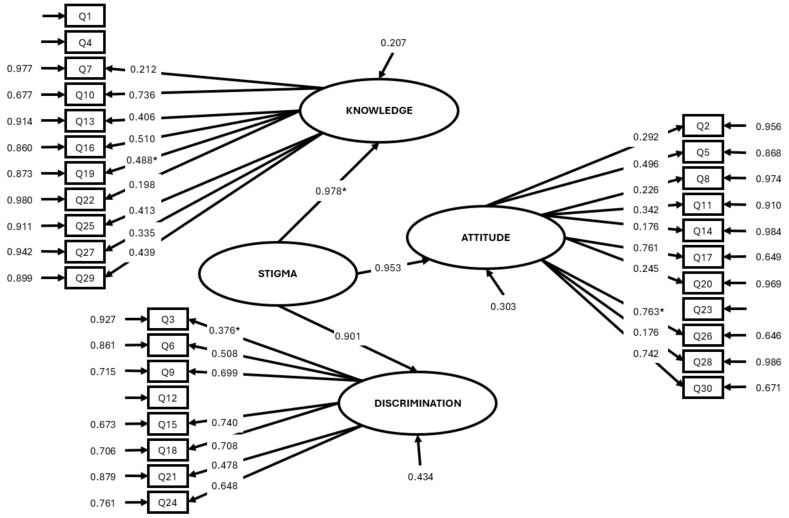
Confirmatory factor analysis model obtained in standardized values. All coefficients are significant. The fixed parameters were marked with “*”.

**Table 1 healthcare-12-01242-t001:** Assessment of social stigma scales and their principal shortcomings.

Scale	Nº Items	Shortcomings
Community Attitudes toward the Mentally Ill (CAMI)—Taylor & Dear, (1981) [26] adpt. sp. by Ochoa et al. (2016) [25]	40	Temporal and cultural decontextualization. Validated in Spanish in school population. Update of concepts, both in understanding and terminology.
Attitudes to Mental Illness Questionnaire (AMIQ)—Cunningham et al. (1993) [22]; Luty et al. (2006) [24]	5	○ Specificity: assessed through 5 questions addressing situations involving: drug use, regular medication consumption, alcohol consumption, criminal behavior, chronic illnesses, religiosity, and severe mental illness.
Beliefs towards Mental Illness scale (BMI)—Hirai & Clum, (2000) [21]	21	Temporal and cultural decontextualization. Lack of conceptual update, both in understanding and terminology. Focused on negative beliefs (dangerousness, deficits in social and interpersonal skills, and incurability).
Day’s Mental Illness Stigma Scale—Day et al. (2007) [23]	28	○ 7 dimensions: interpersonal anxiety, relationship disruption, poor hygiene, visibility, treatability, professional efficacy, and recovery. ○ There are 28 items, but open to consulting any specific mental illness: depression, anxiety, bipolar disorder, schizophrenia, etc. Therefore, the number of items would multiply by the number of mental illnesses under study.
Mental Health Knowledge Schedule (MAKS)—Evans-Lacko et al., (2010) [19]	12	Includes 6 items on general knowledge of mental illness and 6 items on knowledge of individuals with mental health conditions.
Health Styles Survey—Kobau et al. (2010) [29]	11	○ 2 dimensions: negative stereotypes and recovery. Does not follow a theoretical model of stigma associated with mental illness. Developed as a population screening.
Reported and Intended Behavior Scale (RIBS)—Evans-Lacko et al., (2011) [27]	8	Assesses behavior towards individuals diagnosed with mental health conditions in areas of housing, employment, and social relationships, in the present/near past and in the future.
The Stigma-9 Questionnaire (STIG-9)—Gierk et al. (2018) [20]	9	○ Assesses perception about the general population rather than the personal perception of the responding subject. ○ Follows the theoretical model of modified labeling theory (Link et al.) regarding the self-concept of individuals with mental illness. ○ The sample used in the development consists of individuals with some type of mental disorder (affective disorders, anxiety disorders, somatoform disorders, eating disorders, or personality disorders), not the general population.

**Table 2 healthcare-12-01242-t002:** Dimensions of the Spanish S3 initial proposal.

Dimensions	Nº Items	Scoring
Stereotype—Cognitive Dimension.	11	Max.: 55. Min.: 11.
Stigmatizing Attitudes—Affective Dimension.	11	Max.: 55. Min.: 11.
Discrimination—Behavioral Dimension.	8	Max.: 40. Min.: 8.
TOTAL		Max.: 150. Min.: 30.

**Table 3 healthcare-12-01242-t003:** Participant characteristics.

		Frequency	%
Age ($\underline{X},\;Sd)$		35.81	12.53
Gender	Male	161	29.10
	Female	381	68.90
	Other	11	2.00
Ideology	Progressive	309	65.20
	Central	137	28.90
	Conservative	28	5.90
Level of education	Early childhood education	0	0.00
	Primary education	5	0.90
	Compulsory secondary education	19	3.50
	High School	46	8.40
	Vocational training	57	10.40
	University career	222	40.40
	Master’s or postgraduate degree	176	32.00
	PhD	19	3.50
	Other	6	1.10
Contact with mental health	I have no contact with mental health	73	13.00
	I know people with mental health issues	442	78.50
	I am diagnosed with a mental disorder	48	8.50
Training in mental health	Yes, I have training related to mental health	302	54.60
	I have no training related to mental health	251	45.40
Financial situation	Employed	380	69.30
	Self-employed	30	5.50
	Unemployed	116	21.20
	Contributory pension recipient	17	3.10
	Non-contributory pension recipient	0	0.00
	Temporary redundancy plan (ERTE)	5	0.90

**Table 4 healthcare-12-01242-t004:** Item analysis of Spanish S3 dimensions.

Knowledge Dimension α = 0.659, and Ω = 0.655	$\underline{X}$	Sd	r_C_	α_B_	Ω_B_
1. Medication is not an effective treatment for mental health problems.	2.387	1.265	0.120	0.681	0.682
4. People with mental health problems cannot fully recover.	1.851	1.083	0.308	0.638	0.631
7. Psychotherapy is not an effective treatment for people diagnosed with a mental disorder.	1.613	1.075	0.254	0.648	0.649
10. I consider someone who is being treated for mental illness to be dangerous.	1.385	0.799	0.429	0.622	0.617
13. I consider mental illness as a sign of personal weakness.	1.226	0.664	0.324	0.640	0.635
16. (Inv) I consider that people with mental illness are not a danger to others.	1.957	1.276	0.348	0.631	0.623
19. (Inv) It is possible to have a normal relationship with someone with a mental illness.	1.654	1.008	0.361	0.628	0.624
22. Treatments for mental illness are not effective.	1.600	0.958	0.272	0.644	0.645
25. There is little the diagnosed person can do to control the symptoms of mental illness.	1.639	0.915	0.404	0.622	0.617
27. A person who has developed a mental illness may never fully recover from it.	1.750	1.062	0.370	0.626	0.619
29. (Inv) People with mental illness are capable of living on their own.	1.604	0.936	0.350	0.631	0.626
**Attitude Dimension α = 0.660, and Ω = 0.684**	$\underline{X}$	**Sd**	**r_C_**	**α_B_**	**Ω_B_**
2. When a person shows signs of mental disturbance, he/she should be immediately admitted to the hospital.	1.891	1.109	0.181	0.662	0.680
5. People with mental illness should be kept isolated from the community.	1.118	0.471	0.365	0.644	0.667
8. Being part of the community is counterproductive to the recovery of people with mental illness.	1.241	0.738	0.19	0.656	0.681
11. People with mental illness have characteristics that make them easy to distinguish from normal people.	1.703	1.009	0.273	0.645	0.674
14. Greater importance should be given to protecting the population of people with mental illness.	2.617	1.809	0.149	0.707	0.736
17. Living next door to a mentally ill person would scare me.	1.599	0.943	0.592	0.591	0.618
20. (Inv) Treatment can help people with mental illness lead normal lives.	1.307	0.729	0.154	0.661	0.682
23. I am afraid of people with mental illness.	1.466	0.907	0.599	0.592	0.623
26. I worry that, if I am around a mentally disturbed person, I may physically harm myself.	1.622	0.964	0.634	0.579	0.610
28. I am afraid of what they will think of me if I am diagnosed with a mental disorder.	3.247	1.429	0.146	0.682	0.706
30. I am concerned that if I am around a mentally disturbed person, they may verbally assault me.	1.618	1.019	0.616	0.586	0.605
**Discrimination Dimension α = 0.634, and Ω = 0.684**	$\underline{X}$	**Sd**	**r_C_**	**α_B_**	**Ω_B_**
3. (Inv) Living with someone with mental illness is not a problem for me.	2.871	1.340	0.3	0.642	0.708
6. (Inv) I am willing to work with a colleague with mental health issues.	1.468	0.933	0.41	0.578	0.645
9. I avoid working with a mentally ill person as a colleague.	1.381	0.779	0.563	0.540	0.604
12. I can be friends with a person with mental illness.	4.760	0.693	−0.314	0.733	0.740
15. I would avoid crossing paths with the mentally ill if I knew they were in my building.	1.281	0.729	0.585	0.539	0.605
18. I would distance myself from a co-worker if I found out he/she had a mental illness.	1.272	0.698	0.57	0.547	0.610
21. It would force the mentally ill to be admitted to a psychiatric facility.	1.339	0.710	0.305	0.610	0.666
24. I would move if a neighbor had mental illness.	1.158	0.515	0.506	0.582	0.644

r_c_: Item correlation—corrected test; α_B_: Alpha value if the item is deleted; Inv: inverse score.

**Table 5 healthcare-12-01242-t005:** Spanish S3. Corrected scale.

1.	When a person shows signs of mental disturbance, he/she should be immediately admitted to the hospital. (A)	2
2.	(Inv) Living with someone with mental illness is not a problem for me. (D)	3
3.	People with mental illness should be kept isolated from the community. (A)	5
4.	(Inv) I am willing to work with a colleague with mental health issues. (D)	6
5.	Psychotherapy is not an effective treatment for people diagnosed with a mental disorder. (K)	7
6.	Being part of the community is counterproductive to the recovery of people with mental illness. (A)	8
7.	I avoid working with a mentally ill person as a colleague. (D)	9
8.	I consider someone who is being treated for mental illness to be dangerous. (K)	10
9.	People with mental illness have characteristics that make them easy to distinguish from normal people. (A)	11
10.	I consider mental illness as a sign of personal weakness. (K)	13
11.	Greater importance should be given to protecting the population of people with mental illness. (A)	14
12.	I would avoid crossing paths with the mentally ill if I knew they were in my building. (D)	15
13.	(Inv) I consider that people with mental illness are not a danger to others. (K)	16
14.	Living next door to a mentally ill person would scare me. (A)	17
15.	I would distance myself from a co-worker if I discovered he/she had a mental illness. (D)	18
16.	(Inv) It is possible to have a normal relationship with someone with a mental illness. (K)	19
17.	It would force the mentally ill to be admitted to a psychiatric facility. (D)	21
18.	Treatments for mental illness are not effective. (K)	22
19.	I would move if a neighbor had mental illness. (D)	24
20.	There is little the diagnosed person can do to control the symptoms of mental illness. (K)	25
21.	I worry that, if I am around a mentally disturbed person, I may physically harm myself. (A)	26
22.	A person who has developed a mental illness may never fully recover from it. (K)	27
23.	I am afraid of what they will think of me if I am diagnosed with a mental disorder. (A)	28
24.	(Inv) People with mental illness are capable of living alone. (K)	29
25.	I am concerned that if I am around a mentally disturbed person, they may verbally assault me. (A)	30

Inv: Inverse score. K: knowledge dimension (max: 45, min: 9). A: stigmatizing attitude dimension (max: 45, min: 9). D: discrimination dimension (max: 35, min: 7). Total scale max: 125, min: 25. The last column indicates the old numbering of the items.

## Data Availability

The data presented in this study are available on request from the corresponding author due to ethical restrictions.

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
