# Peer review of "Development and Psychometric Properties of the Spanish Social Stigma Scale (S3)"

_healthcare, 2024, doi:10.3390/healthcare12131242_

Round 1

Reviewer 1 Report

Comments and Suggestions for Authors

The text lacks sufficient information on what tools we currently have regarding the stigma of mentally ill people. Such a list of meringues is helpful to indicate what is missing in them and why the Spanish stigma scale had to be developed. There is also a lack of social context regarding the issues discussed and why research in this area may be valuable not only from a scientific perspective, but also from a social one. In general, the text is minimalist, but these may be the editorial guidelines. If not, this part of the article should be clearly enriched. The authors refer to little literature on the subject; the number of cited letters should be increased. especially regarding the title of the article. The literature indicates that the authors are generally sufficiently well versed in the topic. Ad: proposed scale: calculations for the scale have been carried out correctly. However, the subscale related to discrimination makes me a bit dubious. There are a lot of items there that, in my opinion, overlap (6,9,18); this subscale could be tested in this respect.

Comments on the Quality of English Language

The article should undergo thorough linguistic versification; there are sentences in which the meaning is distorted by incorrect combinations of words, e.g. "This refusal to be stigmatized can lead to people with mental health problems... 46"

Author Response

The authors appreciate your time in reviewing our manuscript. Below, we respond to each of your comments.

Comment 1: The text lacks sufficient information on what tools we currently have regarding the stigma of mentally ill people. Such a list of meringues is helpful to indicate what is missing in them and why the Spanish stigma scale had to be developed.

Response 1: The authors thank you for your contribution. We have added a table indicating the most commonly used social stigma assessment scales, highlighting, to our understanding, their deficiencies and what has led us to develop our own scale based on these and the theoretical three-dimensional model (Knowledge or Stereotype - Cognition or Attitude - Behavior or Discrimination) advocated by Michaels et al. (2017) [1], Pla Julián et al. (2013) [14], Reneses et al. (2019 [15] and Pescosolido (2013) [28], among others.

  1. Michaels, P.J.; López, M. Rüsch, N.; Corrigan, P.W. Constructs and concepts comprising the stigma of mental illness. Psychology, Society, & Education. 2017. 4(2), 183. https://doi.org/10.25115/psye.v4i2.490
  2. Pla Julián, I.; Donat, A.A.; Bernabeu Díaz, I. Estereotipos y prejuicios de género: Factores determinantes en Salud Mental. Norte de Salud Mental. 2013. ISSN-e 1578-4940, 46, págs. 20-28.
  3. Reneses, B.; Ochoa, S.; Vila-Badia, R.; Lopez-Micó, C.; Garcia-Andrade, RF.; Rodriguez, R.; Argudo, I.; Carrascosa, C.; Thornicroft, G. Validation of the Spanish version of the discrimination and stigma scale (DISC 12). Actas españolas de psiquiatría. 2019. 47(4), 137-148.
  4. Pescosolido, B.A. The Public Stigma of Mental Illness: What Do We Think; What Do We Know; What Can We Prove? Journal of Health and Social Behavior. 2013. 54(1), 1-21. https://doi.org/10.1177/0022146512471197

Table 1. Assessment of Social Stigma Scales and their Principal Shortcomings.

Additionally, the authors have added the following paragraph, lines 238 and 243:

There is currently no validated scale in Spanish that evaluates social stigma based on the KAB model or three-dimensional model. The available scales are based on other factors such as help-seeking, recognition, support, employment, treatment and recovery, and knowledge about illness, among others [19-21]. Other dimensions investigated are authoritarianism, benevolence, social restrictions, and ideologies regarding mental health [22-26].

Comment 2: There is also a lack of social context regarding the issues discussed and why research in this area may be valuable not only from a scientific perspective, but also from a social one. In general, the text is minimalist, but these may be the editorial guidelines. If not, this part of the article should be clearly enriched. The authors refer to little literature on the subject; the number of cited letters should be increased. especially regarding the title of the article. The literature indicates that the authors are generally sufficiently well versed in the topic.

Response 2: The authors thank you for your contribution. We concur with your point, yet the constraints of the journal have compelled us to be more minimalist (as you suggest) than we would prefer. Perhaps with the table added previously, this issue may also be addressed. Furthermore, we have expanded the discussion by indicating the added value of our scale based on the KCA model compared to previous ones (lines 234 - 243):

In addition, the current scale offers a global measure and a measure of the different spheres of social stigma, offering information on stereotypes, stigmatizing attitudes and discriminatory behaviors and how these three dimensions impact a greater or lesser degree of social stigma.

There is currently no validated scale in Spanish that evaluates social stigma based on the KAB model or three-dimensional model. The available scales are based on other factors such as help-seeking, recognition, support, employment, treatment and recovery, and knowledge about illness, among others [19-21]. Other dimensions investigated are authoritarianism, benevolence, social restrictions, and ideologies regarding mental health [22-26].

Comment 3: Ad: proposed scale: calculations for the scale have been carried out correctly. However, the subscale related to discrimination makes me a bit dubious. There are a lot of items there that, in my opinion, overlap (6,9,18); this subscale could be tested in this respect.

Response 3: The authors thank you for your contribution. While at a content level, the items you mentioned may indeed overlap, statistically, their removal would decrease the consistency of the dimension assessing discrimination. This could be because, as you rightly point out, they correlate with each other, yet in the confirmatory factor analysis, the Wald test did not indicate that covariances between these items should be included to properly fit that dimension. Thus, it leads us to believe that such an association occurs due to the existence of the Discrimination factor. Nevertheless, we could justify that each of these items addresses different conditions of workplace discrimination: willingness to work, avoidance of work, and escape from work with individuals with mental illness.

Comments on the Quality of English Language: The article should undergo thorough linguistic versification; there are sentences in which the meaning is distorted by incorrect combinations of words, e.g. "This refusal to be stigmatized can lead to people with mental health problems... 46"

Response: The authors thank you for your contribution. We have hired a specialized scientific writing service to review the language of the entire manuscript. Edits have been made throughout the text to improve English language, grammar and clarity. Please see the proofreading certificate provided.

As an example, the previously mentioned paragraph was originally worded as follows:

Moreover, the social stigma towards mental illness interferes with the ability for people who begin to suffer from mental health problems and their families to access the specific health and/or social resources necessary to address these problems, due to the lack of identification with the pejorative image of the disorder or the fear of being labeled as being "mentally ill" [7, 11]. This refusal to be stigmatized can lead to people with mental health problems avoiding or refusing to seek professional help [12].

This paragraph has been improved and reworded as shown below, breaking down long sentences and correcting grammatical inconsistencies (lines 41-46):

Moreover, the social stigma towards mental illness interferes with the ability to access the specific health and/or social resources available to those who begin to suffer from mental health problems and their families [7, 11]. Concretely, people may fail to identify with the pejorative image of the disorder or express fear of being labeled as "mentally ill" [7, 11]. This refusal to be stigmatized can lead individuals with mental health problems to avoid or ultimately refuse seeking professional help [12].

Reviewer 2 Report

Comments and Suggestions for Authors

Dear authors: Firstly, I congratulate you on this study you have carried out on the creation of the social stigma scale for mental illnesses or disorders. However, I have some questions that I would like clarified. 1. I have seen that they have not carried out an exploratory factor analysis of the scale. Why haven't they carried out this type of analysis to see the structure of the scale? 2. As a suggestion with the above, a configural invariance analysis could have been carried out with two different samples, one used for the EFA and the other for CFA. 3. Already in the article presented, more specifically in Figure 1, the standardized values ​​of the relationships between the second-order factor with the dimensions do not appear. 4. In table 5, I would like to improve the presentation of the table a little, since it is a bit confusing, with the numbers that appear to know the items that ultimately remain on the scale.

Author Response

Response letter

The authors appreciate your time in reviewing our manuscript. Below, we respond to each of your comments.

Reviewer 2

Comment 1: Dear authors: Firstly, I congratulate you on this study you have carried out on the creation of the social stigma scale for mental illnesses or disorders. However, I have some questions that I would like clarified. 1. I have seen that they have not carried out an exploratory factor analysis of the scale. Why haven't they carried out this type of analysis to see the structure of the scale?

Response 1: The authors thank you for your contribution. Classically (Mulaik, 1972; Matsunaga, 2010), an exploratory factor analysis (EFA) or a confirmatory factor analysis (CFA) is conducted depending on the study's purpose. While both procedures are used to assess the underlying factor structure of the covariance matrix among items, it could be said that EFA is employed to "Create or build" the theory, whereas CFA is used to "confirm" said theory. If we have a limited idea about the construct under study, EFA helps identify the latent factors or dimensions underlying the observed variables (items), as well as the relationships between latent and observed variables. However, if we have a clear model where latent variables are known, and these are used to create observable variables (as is the case at hand), CFA allows us to test the hypothesized factor structure, evaluating whether the proposed model fits the data correctly.

Mulaik, S.A. (1972) The Foundations of Factor Analysis. McGraw-Hill, New York.

Matsunaga, M. (2010). How to Factor Analyze Your Data Right: Do’s, Don’ts, and How-To’s. International Journal of Psychological Research, 3, 97-110.

Comment 2: As a suggestion with the above, a configural invariance analysis could have been carried out with two different samples, one used for the EFA and the other for CFA.

Response 2: The authors thank you for your contribution. We are preparing a second publication once we have the overall validation of the scale, where we will carry out a comparison between subjects who have received specific mental health training (n = 302, 54.6% of our current sample) compared to those who have not. We will include your suggestion to study factorial invariance, however, first we need to have the global validation of the scale. We are very grateful for your suggestion.

Comment 3: Already in the article presented, more specifically in Figure 1, the standardized values ​​of the relationships between the second-order factor with the dimensions do not appear.

Response 3: The authors thank you for your contribution. We apologize for the omission of those loadings. We have included them in the figure. Thank you very much for your guidance.

Comment 4: In table 5, I would like to improve the presentation of the table a little, since it is a bit confusing, with the numbers that appear to know the items that ultimately remain on the scale.

Response 4: The authors thank you for your contribution. We have included a new column with the old item numbering in Table 5. We hope this facilitates the understanding of the structure of the final scale.

Reviewer 3 Report

Comments and Suggestions for Authors

Thank you for opportunity of review. 

Please add the theoretical basis for knowledge, attitudes and behavior, that is, the KAB model, to the introduction.

Suggest a specific research question.

There is no detailed information about the studies reviewed to develop the questions. It is necessary to explain what questions were composed, which questions were used in this study, and whether any questions were modified. It is also necessary to know whether there have been attempts to review foreign literature other than tools or literature developed in Spanish.

Information on the significance of each regression coefficient is needed in the confirmatory factor analysis results.

Although it is said that problematic questions were deleted due to low reliability, there is no visible process of deleting questions that are not significant in the regression coefficient. If the regression coefficients for all questions were significant, please state this.

 Since it appears that the three-factor structure has not been confirmed, the discussion below must be rewritten.

This is appropriate for the measurement of the three specific dimensions that explain the social stigma and contains an adjusted number of items, unlike other previous longer and less complete assessment tools.

 The discussion should be expanded to include the clinical and practical implications of the tools presented as research results and suggestions for future research and limitations.

 The demographic characteristics of survey respondents appear to be biased. There is a need for discussion as to whether this bias will have any effect on the results.

Author Response

Response letter

The authors appreciate your time in reviewing our manuscript. Below, we respond to each of your comments.

Reviewer 3

Comment 1: Thank you for opportunity of review. Please add the theoretical basis for knowledge, attitudes and behavior, that is, the KAB model, to the introduction.

Response 1: We have followed the KAB model throughout the structure of the manuscript, including the theoretical bases. However, we agree with you that we had not named it specifically, we have now named it in several sections, see line 53 (with the theoretical bases between lines 52 and 58), 238-243 and 260-263.

Comment 2: Suggest a specific research question. There is no detailed information about the studies reviewed to develop the questions. It is necessary to explain what questions were composed, which questions were used in this study, and whether any questions were modified. It is also necessary to know whether there have been attempts to review foreign literature other than tools or literature developed in Spanish.

Response 2: The authors thank you for your contribution. We have added a table indicating the most commonly used social stigma assessment scales, highlighting, to our understanding, their deficiencies and what has led us to develop our own scale based on these and the theoretical three-dimensional model (Knowledge or Stereotype - Cognition or Attitude - Behavior or Discrimination) advocated by Michaels et al. (2017) [1], Pla Julián et al. (2013) [14], Reneses et al. (2019 [15] and Pescosolido (2013) [28], among others.

  1. Michaels, P.J.; López, M. Rüsch, N.; Corrigan, P.W. Constructs and concepts comprising the stigma of mental illness. Psychology, Society, & Education. 2017. 4(2), 183. https://doi.org/10.25115/psye.v4i2.490
  2. Pla Julián, I.; Donat, A.A.; Bernabeu Díaz, I. Estereotipos y prejuicios de género: Factores determinantes en Salud Mental. Norte de Salud Mental. 2013. ISSN-e 1578-4940, 46, págs. 20-28.
  3. Reneses, B.; Ochoa, S.; Vila-Badia, R.; Lopez-Micó, C.; Garcia-Andrade, RF.; Rodriguez, R.; Argudo, I.; Carrascosa, C.; Thornicroft, G. Validation of the Spanish version of the discrimination and stigma scale (DISC 12). Actas españolas de psiquiatría. 2019. 47(4), 137-148.
  4. Pescosolido, B.A. The Public Stigma of Mental Illness: What Do We Think; What Do We Know; What Can We Prove? Journal of Health and Social Behavior. 2013. 54(1), 1-21. https://doi.org/10.1177/0022146512471197

Comment 3: Information on the significance of each regression coefficient is needed in the confirmatory factor analysis results.

Although it is said that problematic questions were deleted due to low reliability, there is no visible process of deleting questions that are not significant in the regression coefficient. If the regression coefficients for all questions were significant, please state this.

Since it appears that the three-factor structure has not been confirmed, the discussion below must be rewritten.

Response 3: The authors thank you for your contribution.  The title of the figure indicates that all coefficients were significant, with fixed parameters marked with an "*". The entire procedure has not been included due to space limitations and in keeping with the journal requirements regarding number of tables. Nonetheless, the final structure of the CFA fits correctly, as indicated in the fit indices: χ2SB = 412.0321, gl = 293 (p < .01.), BBNNFI = .922, CFI = .930; IFI = .931; RMSEA = .028 [.022, .035].

Comment 4: This is appropriate for the measurement of the three specific dimensions that explain the social stigma and contains an adjusted number of items, unlike other previous longer and less complete assessment tools. The discussion should be expanded to include the clinical and practical implications of the tools presented as research results and suggestions for future research and limitations.

Response 4: The authors thank you for your contribution. We have added a table indicating the most commonly used social stigma assessment scales, highlighting, to our understanding, their deficiencies and what has led us to develop our own scale based on these and the theoretical three-dimensional model (Knowledge or Stereotype - Cognition or Attitude - Behavior or Discrimination) advocated by Michaels et al. (2017) [1], Pla Julián et al. (2013) [14], Reneses et al. (2019 [15] and Pescosolido (2013) [28], among others.

  1. Michaels, P.J.; López, M. Rüsch, N.; Corrigan, P.W. Constructs and concepts comprising the stigma of mental illness. Psychology, Society, & Education. 2017. 4(2), 183. https://doi.org/10.25115/psye.v4i2.490
  2. Pla Julián, I.; Donat, A.A.; Bernabeu Díaz, I. Estereotipos y prejuicios de género: Factores determinantes en Salud Mental. Norte de Salud Mental. 2013. ISSN-e 1578-4940, 46, págs. 20-28.
  3. Reneses, B.; Ochoa, S.; Vila-Badia, R.; Lopez-Micó, C.; Garcia-Andrade, RF.; Rodriguez, R.; Argudo, I.; Carrascosa, C.; Thornicroft, G. Validation of the Spanish version of the discrimination and stigma scale (DISC 12). Actas españolas de psiquiatría. 2019. 47(4), 137-148.
  4. Pescosolido, B.A. The Public Stigma of Mental Illness: What Do We Think; What Do We Know; What Can We Prove? Journal of Health and Social Behavior. 2013. 54(1), 1-21. https://doi.org/10.1177/0022146512471197

Likewise, we have included the clinical and practical implications of the tools presented as research results and suggestions for future research and limitations in the discussion and we also respond to this comment in the conclusion with information that was already presented in the previous manuscript:

Discussion: Lines 234-243

In addition, the current scale offers a global measure and a measure of the different spheres of social stigma, offering information on stereotypes, stigmatizing attitudes and discriminatory behaviors and how these three dimensions impact a greater or lesser degree of social stigma.

There is currently no validated scale in Spanish that evaluates social stigma based on the KAB model or three-dimensional model. The available scales are based on other factors such as help-seeking, recognition, support, employment, treatment and recovery, and knowledge about illness, among others [19-21]. Other dimensions investigated are authoritarianism, benevolence, social restrictions, and ideologies regarding mental health [22-26].

Conclusion: Lines 256-274

Our tool offers a measure of the three dimensions in a single scale, providing information on the degree of stereotyped thoughts or erroneous knowledge presented by the person, the affective dimension of HS, and rejecting behaviors towards people with a mental health diagnosis, in both a unidimensional and global manner.

In addition, this new social stigma assessment measure provides a useful and rapid measure to be administered in the Spanish population. This is appropriate for the measurement of the three specific dimensions that explain the social stigma, based on KAB model, and contains an adjusted number of items, unlike other previous longer and less complete assessment tools.

Improving our knowledge and understanding of the influence of social stigma towards people with a mental health diagnosis among the Spanish population should serve to detect and respond to current needs and break down the barriers that interfere with the link with Mental Health Associations and Resources. Also, to reinforce the promotion of a holistic and adequate recovery of a person’s objectives in life, promoting a full, meaningful, and satisfactory psychosocial and community rehabilitation for the person with a mental health diagnosis.

In addition, new studies are needed to help expand the quality and number of studies on the importance of social stigma in the general population, and particularly among people diagnosed with mental disorders or "self-stigma”.

Comment 5: The demographic characteristics of survey respondents appear to be biased. There is a need for discussion as to whether this bias will have any effect on the results.

Response 5: The authors thank you for your contribution. We acknowledge that certain demographic characteristics of the sample may influence the results, and have commented on the same in the discussion. However, we would appreciate your insight into what aspects you believe are biasing the results. We hope you are satisfied with the explanations provided in lines 243 to 253:

In this study, it is important to consider the limitations associated with the sample characteristics. Firstly, there is a gender bias, as 68.90% of the participants were women. Another relevant bias is the political bias, given that the predominant ideology in the sample was progressive (65.20%). Likewise, the educational level of the sample (75.90% had university or postgraduate education) could influence attitudes toward social stigma, excluding individuals with lower educational levels. Furthermore, the fact that 54.60% of participants had specific mental health training could bias responses and attitudes. Lastly, prior contact with individuals diagnosed with mental health conditions (78.50% of the sample) could influence perceptions and attitudes toward stigma. In summary, while these findings are valuable, it is essential to acknowledge these limitations and consider how they may affect the generalizability of the results to the overall population.

Round 2

Reviewer 3 Report

Comments and Suggestions for Authors

Thank you for your revision.